# Modelling the Distribution of Cognitive Outcomes for Early-Stage Neurocognitive Disorders: A Model Comparison Approach

**DOI:** 10.3390/biomedicines12020393

**Published:** 2024-02-08

**Authors:** Seyed Ehsan Saffari, See Ann Soo, Raziyeh Mohammadi, Kok Pin Ng, William Greene, Negaenderan Kandiah

**Affiliations:** 1Centre for Quantitative Medicine, Duke-NUS Medical School, National University of Singapore, Singapore 169857, Singapore; raziyeh.mohammadi@duke-nus.edu.sg; 2Department of Neurology, National Neuroscience Institute, Singapore 308433, Singapore; ng.kok.pin@singhealth.com.sg; 3Dementia Research Centre (Singapore), Lee Kong Chian School of Medicine, Nanyang Technological University, Singapore 308232, Singapore; sooseeann@gmail.com (S.A.S.); nagaendran_kandiah@ntu.edu.sg (N.K.); 4Duke-NUS Medical School, National University of Singapore, Singapore 169857, Singapore; 5Stern School of Business (Emeritus), New York University, New York, NY 10012, USA; wgreene@stern.nyu.edu

**Keywords:** cognitive impairment, cognitive screening tool, Montreal Cognitive Assessment, visual cognitive assessment test, skewness

## Abstract

**Background**: Cognitive assessments for patients with neurocognitive disorders are mostly measured by the Montreal Cognitive Assessment (MoCA) and Visual Cognitive Assessment Test (VCAT) as screening tools. These cognitive scores are usually left-skewed and the results of the association analysis might not be robust. This study aims to study the distribution of the cognitive outcomes and to discuss potential solutions. **Materials and Methods**: In this retrospective cohort study of individuals with subjective cognitive decline or mild cognitive impairment, the inverse-transformed cognitive outcomes are modelled using different statistical distributions. The robustness of the proposed models are checked under different scenarios: with intercept-only, models with covariates, and with and without bootstrapping. **Results**: The main results were based on the VCAT score and validated via the MoCA score. The findings suggested that the inverse transformation method improved the modelling the cognitive scores compared to the conventional methods using the original cognitive scores. The association of the baseline characteristics (age, gender, and years of education) and the cognitive outcomes were reported as estimates and 95% confidence intervals. Bootstrap methods improved the estimate precision and the bootstrapped standard errors of the estimates were more robust. Cognitive outcomes were widely analysed using linear regression models with the default normal distribution as a conventional method. We compared the results of our suggested models with the normal distribution under various scenarios. Goodness-of-fit measurements were compared between the proposed models and conventional methods. **Conclusions**: The findings support the use of the inverse transformation method to model the cognitive outcomes instead of the original cognitive scores for early-stage neurocognitive disorders where the cognitive outcomes are left-skewed.

## 1. Introduction

The recognition of cognitive impairment in neurocognitive disorders is important due to its association with shorter life expectancy, caregiver anxiety, and the potential side-effects of cognitive enhancers. Furthermore, early diagnosis and timely intervention can result in the delayed or slowed progression from mild cognitive impairment (MCI) to dementia. Hence, it is paramount to detect early cognitive deficits to provide appropriate treatment decisions to reduce complications and morbidities. For these reasons, routine cognitive screening is important for the optimal management of patients with neurocognitive disorders [1,2,3,4,5,6,7,8]. Though the gold standard of such cognitive evaluation should be based on neuropsychological testing, these are time-consuming, and hence a more practical screening approach is usually performed by clinicians. Cognitive tests are usually performed to assess patients in clinical conditions with cognitive impairment. Such cognitive tests should be short, simple to perform, sensitive to subtle changes in cognition, not confounded by motor and visual problems, and able to assess a full range of cognitive domains. Some global cognitive screening tools that have been developed for patients with neurocognitive disorders are the Montreal Cognitive Assessment (MoCA) and Visual Cognitive Assessment Test (VCAT), among others [4,7,9,10,11,12,13].

Among the screening tools, the MoCA is a screening instrument that can be administered within 10–15 min and is developed to identify patients with MCI and dementia [13,14]. The VCAT is a visual-based cognitive screening tool that is comparable to the MoCA for the time of administration, is designed to detect early cognitive impairment, and takes 15.7 ± 7.3 min to complete [11,12,15]. The MOCA and VCAT have been demonstrated to be particularly useful for the detection of early-stage neurocognitive disorders, such as MCI and mild dementia. The outcome for each of these screening tools total up to 30 points and range in integers between 0 and 30, where a lower score indicates worse cognition. In cross-sectional and association studies, these outcomes are usually treated as continuous variables. Therefore, in terms of statistical analysis, linear regression models are likely performed to study the outcomes of the MoCA and VCAT [11,16,17].

There are some limitations while using linear regression models to examine the outcomes of such screening tools. One such limitation is that the underlying statistical distribution of the residuals is assumed to be normally distributed. While this assumption can be checked using a normal quantile (Q-Q) plot, histogram, or goodness-of-fit test (e.g., the Kolmogorov–Smirnov test), it should be noted that such goodness-of-fit tests lack sensitivity in small studies, whereas, on the other hand, the violation of the normality assumption becomes less important with increasing sample sizes. In addition, a major limitation of the normal distribution for the cognitive outcomes is that the predicted values are symmetric around the mean value. This is usually not the case for left-skewed outcome variables, such as the MoCA and VCAT, when used to detect the early stages of neurocognitive disorders where a high proportion of individuals are on the right side of the distribution and minorities are on the left side. Outcome transformation is one solution to overcome the skewness. Such transformation should be straightforward and generate interpretable results. However, the results of a complex transformation may not be transparent and need back-calculations [18,19].

To counter the limitations of the normal distribution in linear regression models, which can potentially provide predicted negative values, some alternative statistical distributions may be worth considering to overcome this problem and eventually improve the model performance and predictions. However, the domain of some statistical distributions is for non-negative values, and, as mentioned above, the cognitive outcomes are non-negative integers (ranged between 0 and 30), which seems to fit such non-negative outcomes more appropriately. In addition, count distributions belong to this category, and they have the advantages of dealing with integers. Given that the two cognitive outcomes are right-skewed, an alternative option is to look at the inverse outcomes (which is left-skewed) and fit statistical distributions where they allow for left-skewness. This paper aims to investigate the distribution of the cognitive outcomes under various scenarios. First, both the VCAT outcome and inverse VCAT outcome will be modelled using different statistical distributions in the presence and absence of predictor variables. Next, the bootstrap method will be applied to investigate whether such resampling techniques could improve the modelling. Lastly, goodness-of-fit measurements will be compared among the candidate models. The results of the various scenarios mentioned above will be validated using the MoCA cognitive outcome and its inverse version.

## 2. Materials and Methods

### 2.1. Study Setting

In this retrospective cohort study, 883 participants at risk of dementia, having a diagnosis of subjective cognitive decline (SCD) or MCI, were recruited from the community as well as from a tertiary neurology centre between January 2015 and December 2020 in Singapore. These participants were recruited from the community through posters; the participants reported subjective cognitive complains and were not diagnosed with dementia at the time of recruitment. This study was approved by the SingHealth Centralized Institutional Review Board and written informed consent was obtained according to the Declaration of Helsinki from all participants. Participants with SCD had symptoms of cognitive impairment, but did not have objective cognitive deficits and were functionally intact. Participants with MCI had symptoms of cognitive impairment and objective cognitive deficits on testing; however, they remained functionally intact as per the Diagnostic and Statistical Manual of Mental Disorders, Fifth Edition (DSM-5), criteria [20] or the National Institute on Aging-Alzheimer’s Association (NIA-AA) criteria [21]. Participants were excluded if they had a prior diagnosis of major psychiatric diseases or dementia.

### 2.2. Outcome Assessments

Data on demographics (including age, gender, and years of education) and cognitive outcomes (VCAT and MoCA) were obtained through clinical interviews and assessments. The two cognitive outcomes in this paper were the VCAT and MoCA.

The MoCA is a widely used cognitive screening test, and its items include attention, concentration, executive functions, memory, language, visuospatial skills, abstraction, calculation, and orientation [10,22]. The MoCA adds one point for those whose educational level is ≤12 years or <10 years in the local population [23]. More recently, the VCAT was developed as a language-neutral cognitive screening tool to detect early cognitive impairment. The main advantage of the VCAT is that there is no necessity to translate the test as long as the rater and the participant speak the same language; thus, it is more applicable to multilingual populations throughout the world. This visual-based test includes 11 items, which contains 5 specific cognitive domains: episodic memory, attention/working memory (WM), executive function, visuospatial ability, and language [12,15].

### 2.3. Outcome Distribution

The two cognitive outcomes (VCAT and MoCA) are scored from 0 to 30, whereby a higher score denotes higher cognitive functioning. As can be seen in Figure 1, they are left-skewed. The distribution of the inverse outcomes (indicating the number of errors in the questionnaire), where a lower score indicates higher cognitive functioning, is shown in Figure 1. The inverse outcomes are also scored between 0 and 30, and defined as follows:(1)$$\begin{array}{ccc} Inverse\left(MoCA\right) & = & 30-MoCA\\ Inverse\left(VCAT\right) & = & 30-VCAT \end{array}$$

### 2.4. Statistical Analysis Plan

The distribution of cognitive outcomes is modelled using the normal distribution as the baseline model. Although normal distribution is considered mostly as the default model, the distribution of cognitive outcomes is not symmetric and some skewness is usually observed. To address this issue, log-normal, gamma, and Weibull distributions are used as potential candidate models. As the cognitive outcome variables (including the VCAT and MoCA) can only be non-negative integers, count distributions were also considered as alternative models. Count models considered in this paper include Poisson, negative binomial (NB), Conway–Maxwell–Poisson (CMP), and the generalised Poisson (GP).

The two cognitive outcomes (original scores) are left-skewed; hence, their inverse score would be right-skewed, as seen in Figure 1. As most of the above distributions could handle right-skewed data, the same models are used for the inverse outcomes (right-skewed) as well. The proposed alternative distributions have shape and scale parameters, which makes them flexible towards skewness. Hence, the choice of the above models was based on the distributional specifications of the cognitive outcomes.

The normal distribution along with the proposed distributions were modelled to the cognitive outcomes and their inverse versions. For each model, two scenarios were explored: the intercept-only model (without predictor variables) and models with predictors (age, gender, and education). Intercept-only models were explored to purely investigate the distribution of the cognitive outcomes, whereas the models with predictors were studied to improve the fitted distributions and explore the associations of the baseline characteristics and cognitive outcomes.

The bootstrap resampling method was applied to investigate whether bootstrapping could improve the model performance under the above two scenarios (intercept-only and models with predictors) and for both the VCAT and inverse (VCAT) outcomes. The unrestricted random method was used as a resampling technique, which selects patients with equal probability and with replacement; therefore, patients could be selected for the bootstrapped sample more than once. The sampling rate of 100% was used, where the results in bootstrapped sample sizes of 883 (which equals to the cohort sample size) and 10,000 replications were used. The bootstrap percentile method was used to calculate the confidence interval (CI). A 95th percentile bootstrap CI with 10,000 bootstrap samples was determined using the interval between the 2.5th percentile and the 97.5th percentile value of the 10,000 bootstrap parameter estimates, i.e., the minimum and maximum value of the 10,000 bootstrap estimates after excluding the 2.5% in each tail of the distribution.

Model performance was compared using goodness-of-fit statistics, including the −2log-likelihood (−2LL), Akaike’s information criterion (AIC), and Bayesian information criterion (BIC) measurements. Predicted VCAT outcomes were compared to the observed VCAT for all the above settings. To validate the findings on the model performance of the VCAT outcome, a similar methodology was applied to the MoCA outcome. Goodness-of-fit statistics and model fits were compared across the two cognitive outcomes. Statistical analysis was performed using PROC NLMIXED in SAS software version 9.4 for Windows (SAS Institute Inc., Cary, NC, USA).

## 3. Results

### 3.1. Patient Characteristics

Descriptive statistics of the demographic variables are reported in Table 1. In this cross-sectional study, 883 patients are enrolled. The mean age is 63.5 ± 7.8 years, with 302 (34.2%) being male. The median education is 11 years (first- and third-quartile of 10–14 years). The mean/median of the VCAT and MoCA are as follows: 24.6/26 and 25.6/27. The distribution of the cognitive outcomes is shown in Figure 1. Left-skewness with a long tail on the left side of the distribution is observed in the VCAT and MoCA scores. The inverse outcomes, on the other hand, show a long tail on the right side of the distribution. Mean and median values of the outcome and inverse outcome scores, shown in Figure 1, indicate that a normal distribution may not be a good model to fit the data.

### 3.2. Model Comparisons—VCAT

Goodness-of-fit measurements are reported in Table 2. A lower −2LL (as well as AIC and BIC) value indicates a better fit. Normal distribution outperforms all other alternative models, except the Weibull distribution, in both the intercept-only models and the models with covariates. The predicted model-based values for the VCAT using the Weibull model also results in a better fit compared to all other distributions (Figure 2). It should also be noted that the Poisson model shows a similar fit compared to the CMP and GP under the intercept-only models and the models with covariates.

### 3.3. Model Comparisons—Inverse VCAT

For the inverse VCAT, the GP and NB models, followed by the gamma model, show better performance in terms of the goodness-of-fit measurements under the intercept-only scenario. In the models with covariates, the GP and gamma distributions, followed by the NB model, outperform the other models (Table 3). The same results could be extracted from the predicted VCAT values by looking at Figure 3. It should also be noted that the log-normal model overestimates the predicted inverse VCAT values at the left side of the distribution (when the original VCAT values are high), and the fitted values out of the normal distribution are far off from the observed values and the other models’ predictions. The model goodness-of-fits also indicate that the Poisson model results in the highest statistics, i.e., the poorest model.

### 3.4. Bootstrapping Method

The models show similar results using the bootstrap method. The Weibull model outperforms other models under the intercept-only model and the models with predictors when looking at the VCAT outcome (Table 4). For the inverse VCAT outcome, the GP, NB, and gamma models indicate better performance compared to the other distributions (Table 5). Similar conclusions could be made based on the predicted values for the VCAT and inverse VCAT scores in Figure 4.

### 3.5. Association Analysis

Using the model with the best-fit performances, the association analyses of the baseline characteristics and VCAT outcomes were reported as beta coefficients and 95% confidence intervals using the Weibull model. Male gender and lower education were found to be significantly associated with worse VCAT scores with and without bootstrapping. For the inverse VCAT outcome, the association results using the GP model (as the best model) are reported. Older age and lower education were found to be significantly associated with worse VCAT scores. The results of the VCAT and inverse VCAT models are not consistent for age (being significant in the inverse VCAT model but not in the VCAT model) and gender (being significant in the VCAT model but not in the inverse VCAT). Except for gender variables under the VCAT model, the confidence intervals are very similar to the bootstrapped confidence intervals (Table 6).

### 3.6. Validation of the Findings

The findings of the model performances on the VCAT scores are validated using the MoCA outcome scores. The Weibull model (followed by the normal distribution) consistently shows the lowest goodness-of-fit measurements under the intercept-only models and the models with covariates in the MoCA outcome. In the inverse outcome scenarios (for the MoCA), the log-normal followed by GP and NB (and then gamma) models indicate the best performance based on the goodness-of-fit statistics (Appendix A). The predicted scores of the MoCA, and their inverse scores, also result in similar findings (Appendix A).

## 4. Discussion

This study investigated the distribution of cognitive scores in the early stages of neurocognitive disorders in different statistical models. Different models (including normal, log-normal, gamma, Poisson, negative binomial, Conway–Maxwell–Poisson, generalised Poisson, and Weibull) were conducted on the VCAT outcome as the primary endpoint variable. The choice of the above distributions was based on the distribution of the original cognitive outcomes and the inverse-transformed outcomes, where these models can potentially provide a better fit performance in the presence of skewness in the outcome variable. Initially, some simulation studies were performed to evaluate the fit performance of each model separately, assuming the same distribution for the outcome variable in the simulation part. It was found that all the proposed distributions showed promising performance in terms of the parameter estimates, as well as the goodness-of-fit measurements, and fit the simulated data fairly robustly, confirming the choice of such models to investigate further.

Intercept-only models and models with covariates were considered, and the model fits were compared using goodness-of-fit measurements. Bootstrapping methods were conducted to calculate the robust confidence intervals for the point estimates. The inverse VCAT outcome was modelled using the same distributions and under the above scenarios (the intercept-only models and the models with covariates, with and without bootstrapping) to present the precision of the estimates via confidence intervals. To validate the findings, similar methods were applied to the MoCA outcome variable.

Our findings showed that the Weibull distribution is a better fit in both the intercept-only models and models with covariates on the VCAT compared to the other models. This is because the other models are not flexible towards left-skewed data, while the Weibull distribution could deal with such data. An alternative model to consider is normal distribution, which showed a better fit compared to the other models. This may be because the other distributions were slightly right-skewed models and away from being symmetric. Hence, the normal distribution (symmetric model) will be preferred over right-skewed models with the left-skewed VCAT outcome.

Under the inverse VCAT outcome scenario, the GP model indicated a better fit compared to the other distributions for both the intercept-only models and the models with covariates. In addition, the NB and gamma models also indicated a close fit to the GP model. This is an expected finding, given that the GP, NB, and gamma models can handle right-skewed distributions. However, at the left side of the distribution (within the cognitively normal range), all the above three models showed a reasonably close fit to the observed scores, where the Weibull and normal distributions underpredicted and the log-normal model overpredicted the outcome. It is also noted that all the above models performed better under the inverse VCAT compared to the original VCAT results, unlike the normal distribution, which showed a similar fit under both scenarios, which was expected, as such an outcome transformation does not reduce the data skewness, i.e., only switches the left-skewness to right-skewness, where both are far away from a symmetric distribution.

Among the methods with the bootstrap method, similar findings were found under both the VCAT (the Weibull model outperformed the other models) and the inverse VCAT (with the GP followed by the NB and gamma models showing the best fit) scenarios. Comparing the goodness-of-fit statistics between the models with and without bootstrapping is not recommended, because such statistics are used as model-selection criteria among the various distributions under the same settings. Bootstrap methods improved the estimated precision and the bootstrapped standard errors of the estimates were more robust. Hence, it is recommended to use bootstrapping methods to present robust estimates along with a 95% CI for both the VCAT and inverse VCAT models for all models.

The results of the VCAT and inverse VCAT modelling were validated on the MoCA outcome variable. As the distribution of the MoCA scores is right-skewed, the Weibull model, as expected, showed a better fit compared to the other models. Hence, the findings are consistent across the two cognitive outcomes under the original scores. However, when using the inverse score method, the results of the model fits for the MoCA are different compared to the VCAT outcome. The log-normal distribution indicated the lowest goodness-of-fit statistics compared to the other models for the inverse MoCA outcome (Appendix A). This is an inconsistent finding compared to the results of the inverse VCAT outcome, where the three other distributions (GP, NB, and gamma) showed a better performance compared to the log-normal model. Looking at the distribution of the cognitive outcomes (Figure 1), the MoCA is a unimodal distribution, unlike the VCAT, with a bimodal distribution. Hence, it seems that the GP, NB, and gamma models are not flexible enough to handle bimodal distributions, and the log-normal model could fit bimodal data better than alternative distributions. Another point which should not be ignored is the good fit performance of the log-normal distribution under the inverse MoCA models at a normal range of cognition, where the GP, NB, and gamma models underestimated the scores. Normal distribution, under the inverse scenario, showed the poorest fit for the two cognitive inverse outcomes (Appendix A).

The three predictor variables (age, gender, and education) were selected, as they are clinically relevant predictors of cognitive outcomes. It should be noted that the cognitive outcomes are the total raw scores and not adjusted for the age or education level. The results of the association analysis indicated that male gender and lower education, but not age, are significantly associated with worse VCAT outcomes using the Weibull distribution. Same findings are seen when testing the MoCA as the cognitive outcome. On the other hand, we found inconsistent findings when testing the two cognitive outcomes using the inverse score scenarios in terms of the significance of the predictor variables. When using the GP model, we found that higher age and lower education are significantly associated with worse VCAT outcomes, while male gender was marginally associated with a worse VCAT outcome. However, male gender and lower education were significantly associated with a worse MoCA outcome under the GP model and the inverse transformation scenario (Appendix A). Although the association directions (positive/negative association) were consistent across all models and scenarios (except the Weibull model under the inverse-transformed outcomes), and make clinical sense, it should be noted that the effect sizes might be different. Hence, a wrongly assumed distribution for cognitive outcomes may under- or overestimate the effect size of the predictors or the statistical significance. For simplicity, other variables (such as family history, neuroimaging findings, etc.) are not included in this paper; however, the conclusion in terms of the choice of the model distribution and inverse transformation will remain the same.

As the original cognitive scores are left-skewed, there might be some computational issues with models without distributional flexibility towards left-skewness. For example, the final Hessian matrix may be full rank, but has at least one negative eigenvalue, or the second-order optimality condition may be violated. Not-estimable confidence intervals reported in Appendix A are a few examples of such computational issues, where the Moore–Penrose inverse is used in the covariance matrix and the estimates are reported based on the finite difference approximation used for the derivative of the probability density function. However, while there are potential solutions to this issue (such as changing the optimisation technique or the integration method), it should be noted that a reasonable model is selected to apply. On the other hand, under the inverse transformation scenario, no computational and convergence problem was found.

The results of this study support the use of the analysis of the inverse cognitive scores over the original scores when comparing the results of the inverse cognitive scores versus the original scores in terms of the model performance of each distribution. From a clinical point of view, the inverse score actually measures the number of errors corresponding to the specific cognitive outcome of interest. That means higher cognitive functioning (a greater value in the original cognitive outcome) equals to a lower number of errors (a smaller value of the inverse cognitive outcome). Hence, the results of the suggested inverse transformation are easy to interpret and are transparent compared to the other potential transformations. An example of a challenging transformation, where a back-calculation is required to interpret the results, is the square root of the number of errors, which was suggested to reduce the bias (to justify the normality assumption of the normal distribution) [18,19]. Compared to such an alternative transformation, our suggested solution is simple, robust, and easy to interpret.

The limitations of this study include the limited number of models performed. More complex distributions can be applied on cognitive outcomes and the results could be compared to the results of this study. Although this study tried to validate the results via an additional cognitive outcome with a similar distributional pattern, validation on the independent cohorts will be needed to verify the findings.

## 5. Conclusions

The findings of this study suggest that the inverse transformation method improves the modelling of the cognitive scores for all proposed models, except the normal distribution. Therefore, the recommendation is to use the inverse cognitive outcomes instead of the original cognitive scores, no matter how the outcome distribution looks (unimodal/bimodal) at the early stage of the disease, where the cognitive outcomes are left-skewed. The proposed inverse method and the suggested statistical distributions could also be applied to other behavioural outcomes where the original scores are left-skewed (such as Mini-Mental State Examination, MMSE); however, one should be careful with the outcome distribution. Although the normal distribution is commonly used for cognitive outcomes, the results of this study show that the model performance of normal distribution is the poorest among the proposed inverse models. Under the unimodal scenario, it is recommended to model the inverse cognitive outcomes via the log-normal distribution and GP distribution (as the alternative model). However, the GP and NB distributions (and gamma as the alternative model) are recommended in the presence of bimodal inverse cognitive variables. When applying the above-proposed models on inverse cognitive outcomes, it is suggested that the model performances should be assessed via predicted scores versus observed cognitive scores as well as goodness-of-fit statistics to make sure the findings are valid and robust. In conclusion, future studies can consider using inverse models to analyse the outcomes of cognitive screening tools, such as the MoCA and VCAT.

## Figures and Tables

**Figure 1 biomedicines-12-00393-f001:**
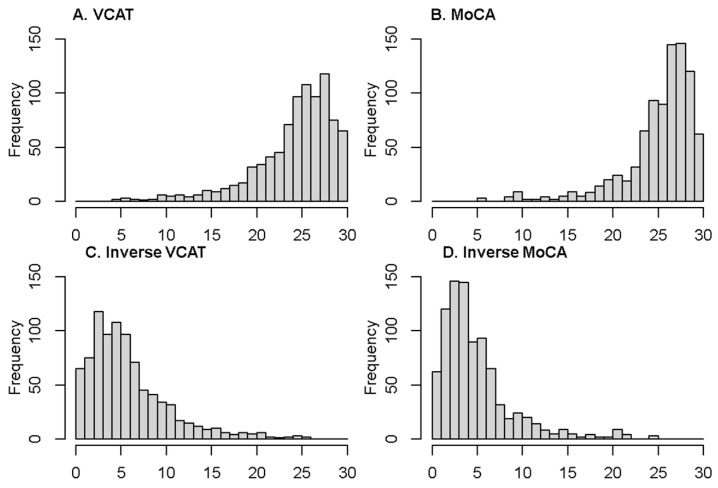
Distribution of the two outcomes and their inverse version (VCAT and MoCA).

**Figure 2 biomedicines-12-00393-f002:**
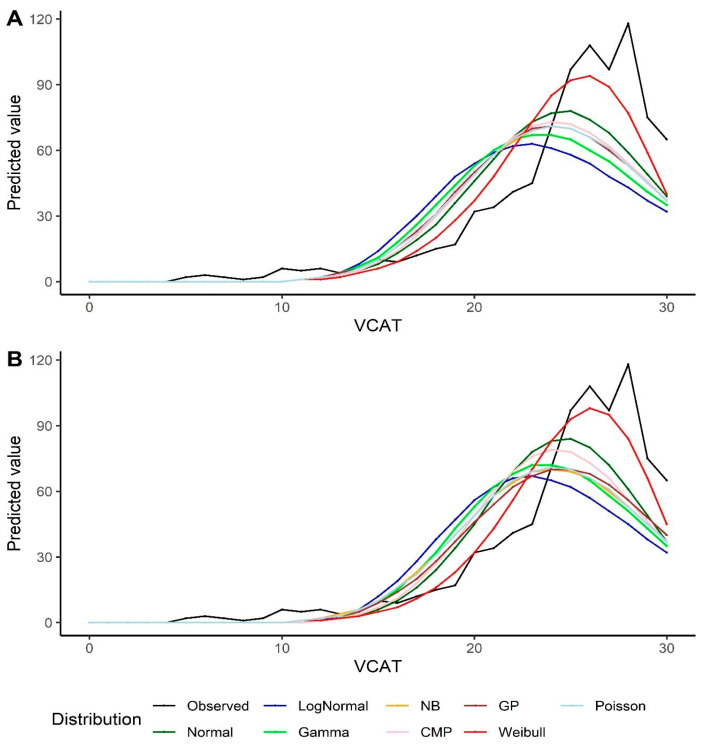
Predicted values by various distributions under different scenarios—VCAT outcome: (**A**) Without covariates (intercept-only); (**B**) With covariates (age, gender, and education variables as predictors).

**Figure 3 biomedicines-12-00393-f003:**
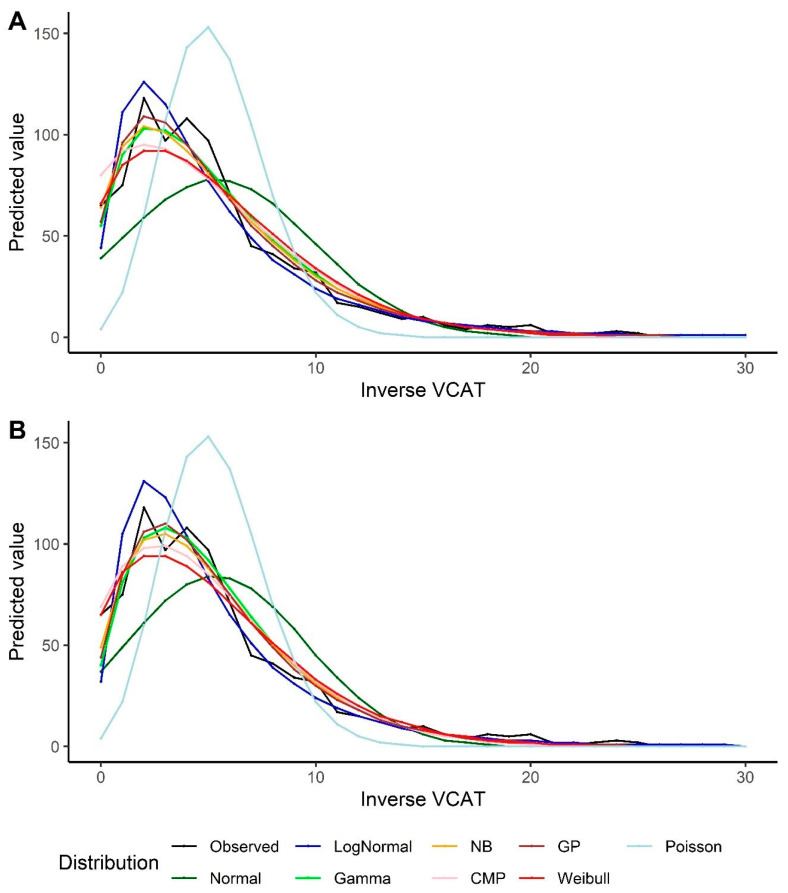
Predicted values by various distributions under different scenarios—VCAT inverse outcome: (**A**) Without covariates (intercept-only); (**B**) With covariates (age, gender, and education variables as predictors).

**Figure 4 biomedicines-12-00393-f004:**
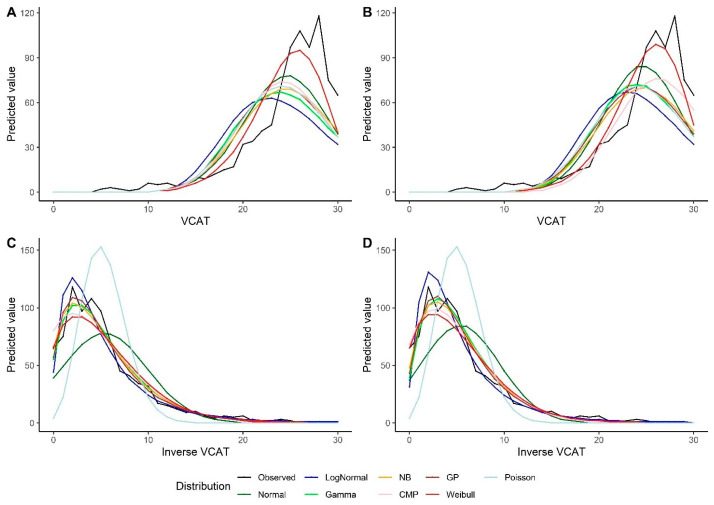
Predicted values by various distributions under different scenarios—VCAT and inverse VCAT outcomes with the bootstrap: (**A**) Without covariates (intercept-only), original score; (**B**) With covariates (age, gender, and education variables as predictors), original score; (**C**) Without covariates (intercept-only), inverse score; (**D**) With covariates (age, gender, and education variables as predictors), inverse score.

**Table 1 biomedicines-12-00393-t001:** Patients’ characteristics and outcome distributions, n = 883.

Variable	Frequency (%)	Mean ± SD	Median (Q1–Q3)
Age (year)		63.5 ± 7.8	63 (58–69)
Male Gender	302 (34.2%)		
Education (year)		11.6 ± 4	11 (10–14)
MoCA		25.6 ± 4.1	27 (24–28)
VCAT		24.6 ± 4.5	26 (23–28)

Abbreviations: MoCA = Montreal Cognitive Assessment; VCAT = Visual Cognitive Assessment Test; SD = standard deviation; Q1 and Q3 = first and third quartile.

**Table 2 biomedicines-12-00393-t002:** Model performance—VCAT outcome.

Distribution	Intercept-Only Model			Model with Covariates		
	−2LL	AIC	BIC	−2LL	AIC	BIC
Normal	5170.6	5174.6	5184.2	5028.6	5038.6	5062.5
Log-Normal	5603.5	5607.5	5617.0	5490.4	5500.4	5524.3
Gamma	5432.7	5436.7	5446.3	5304.3	5314.3	5338.2
Poisson	5279.9	5281.9	5286.7	5165.4	5173.4	5192.6
NB	5415.4	5419.4	5429.0	5452.8	5462.8	5486.7
CMP	5278.1	5282.1	5291.6	5146.8	5156.8	5180.7
GP	5280.2	5284.2	5293.8	5171.7	5181.7	5205.6
Weibull	4992.9	4996.9	5006.5	4916.8	4926.8	4950.7

Abbreviations: LL = log-likelihood; AIC = Akaike’s information criterion; BIC = Bayesian information criterion; NB = negative binomial; CMP = Conway–Maxwell–Poisson; GP = generalised Poisson.

**Table 3 biomedicines-12-00393-t003:** Model performance—VCAT inverse outcome.

Distribution	Intercept-Only Model			Model with Covariates		
	−2LL	AIC	BIC	−2LL	AIC	BIC
Normal	5170.6	5174.6	5184.2	5028.6	5038.6	5062.5
Log-Normal	4791.9	4795.9	4805.4	4654.2	4664.2	4688.1
Gamma	4780.9	4784.9	4794.5	4634.5	4644.5	4668.4
Poisson	5878.0	5880.0	5884.8	5459.4	5467.4	5486.5
NB	4770.1	4774.1	4783.6	4646.6	4656.6	4680.6
CMP	4786.9	4790.9	4800.5	4668.2	4678.2	4702.2
GP	4766.7	4770.7	4780.3	4626.2	4636.2	4660.1
Weibull	4817.1	4821.1	4830.7	4801.0	4811.0	4834.9

Abbreviations: LL = log-likelihood; AIC = Akaike’s information criterion; BIC = Bayesian information criterion; NB = negative binomial; CMP = Conway–Maxwell–Poisson; GP = generalised Poisson.

**Table 4 biomedicines-12-00393-t004:** Model performance—VCAT outcome with the bootstrap.

Distribution	Intercept-Only Model			Model with Covariates		
	−2LL	AIC	BIC	−2LL	AIC	BIC
Normal	5167.33 * (5169.01)	5171.33 (5173.01)	5180.89 (5182.58)	5020.69 (5022.41)	5030.69 (5032.41)	5054.6 (5056.33)
Log-Normal	5596.42 (5599.39)	5600.42 (5603.39)	5609.98 (5612.96)	5477.49 (5480.48)	5487.49 (5490.48)	5511.41 (5514.39)
Gamma	5427.36 (5429.73)	5431.36 (5433.73)	5440.93 (5443.3)	5292.85 (5294.5)	5302.85 (5304.5)	5326.76 (5328.42)
Poisson	5279.28 (5277.90)	5281.28 (5279.90)	5286.06 (5284.69)	5161.84 (5160.15)	5169.84 (5168.15)	5188.97 (5187.29)
NB	5285.56 (5284.23)	5289.56 (5288.23)	5299.12 (5297.80)	5875.61 (5182.70)	5885.61 (5192.70)	5909.52 (5216.62)
CMP	5274.06 (5275.76)	5278.06 (5279.76)	5287.62 (5289.32)	5138.46 (5140.29)	5148.46 (5150.29)	5172.37 (5174.2)
GP	5279.88 (5278.5)	5283.88 (5282.5)	5293.45 (5292.07)	5174.34 (5171.94)	5184.34 (5181.94)	5208.25 (5205.85)
Weibull	4990.21 (4991.17)	4994.21 (4995.17)	5003.78 (5004.74)	4909.14 (4909.17)	4919.14 (4919.17)	4943.06 (4943.08)

* Reported numbers are the mean (median). Abbreviations: LL = log-likelihood; AIC = Akaike’s information criterion; BIC = Bayesian information criterion; NB = negative binomial; CMP = Conway–Maxwell–Poisson; GP = generalised Poisson.

**Table 5 biomedicines-12-00393-t005:** Model performance—VCAT inverse outcome with the bootstrap.

Distribution	Intercept-Only Model			Model with Covariates		
	−2LL	AIC	BIC	−2LL	AIC	BIC
Normal	5167.33 * (5169.01)	5171.33 (5173.01)	5180.89 (5182.58)	5020.69 (5022.41)	5030.69 (5032.41)	5054.6 (5056.33)
Log-Normal	4789.75 (4790.15)	4793.75 (4794.15)	4803.32 (4803.72)	4648.85 (4649.1)	4658.85 (4659.1)	4682.77 (4683.02)
Gamma	4778.83 (4779.02)	4782.83 (4783.02)	4792.4 (4792.59)	4629.23 (4629.44)	4639.23 (4639.44)	4663.15 (4663.35)
Poisson	5874.34 (5873.37)	5876.34 (5875.37)	5881.12 (5880.16)	5444.91 (5444.17)	5452.91 (5452.17)	5472.04 (5471.31)
NB	4767.86 (4767.99)	4771.86 (4771.99)	4781.43 (4781.55)	4626.65 (4627.04)	4636.65 (4637.04)	4660.56 (4660.96)
CMP	4784.53 (4784.92)	4788.53 (4788.92)	4798.1 (4798.48)	4661.92 (4662.69)	4671.92 (4672.69)	4695.83 (4696.61)
GP	4764.55 (4764.88)	4768.55 (4768.88)	4778.12 (4778.45)	4620.38 (4620.63)	4630.38 (4630.63)	4654.29 (4654.55)
Weibull	4814.9 (4815.09)	4818.9 (4819.09)	4828.47 (4828.66)	4795.86 (4796.39)	4805.86 (4806.39)	4829.77 (4830.31)

* Reported numbers are the mean (median).

**Table 6 biomedicines-12-00393-t006:** Association analysis of the baseline characteristics with VCAT and inverse (VCAT) outcomes.

Variable	Whole Cohort (n = 883)		Whole Cohort (n = 883) + Bootstrap Method	
	Beta (95% CI)	*p* Value	Beta (95% CI)	*p* Value
VCAT				
Age (year)	−0.31 (−0.75, 0.13)	0.162	−0.33 (−1.07, 0.35)	0.261
Female Gender	1.47 (0.67, 2.26)	<0.001	1.46 (0.36, 2.53)	0.037
Education (year)	1.05 (0.87, 1.24)	<0.001	1.05 (0.85, 1.26)	<0.001
Inverse (VCAT)				
Age (year)	0.15 (0.10, 0.20)	<0.001	0.15 (0.09, 0.21)	<0.001
Female Gender	−0.10 (−0.20, 0.01)	0.062	−0.10 (−0.20, 0.01)	0.174
Education (year)	−0.19 (−0.22, −0.15)	<0.001	−0.19 (−0.23, −0.15)	<0.001

Abbreviations: VCAT = Visual Cognitive Assessment Test; CI = confidence interval.

## Data Availability

All data and materials are available from the corresponding author upon reasonable request.

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
