# Peer review of "Modelling the Distribution of Cognitive Outcomes for Early-Stage Neurocognitive Disorders: A Model Comparison Approach"

_biomedicines, 2024, doi:10.3390/biomedicines12020393_

Round 1
Reviewer 1 Report
Comments and Suggestions for Authors
In the submitted manuscript, the authors evaluate several statistical models to describe the results of cognitive screening of adults at risk for neurocognitive disorders such as Alzheimer’s. The authors focused on two such screening tools: the Montreal Cognitive Assessment (MoCA) and the Visual Cognitive Assessment Test (VCAT) and evaluated various statistical models to evaluate the impact of cognitive scores. Overall, there are details in the manuscript that need to be clarified.
General Comments:
Contradictions or grammatical errors?
Some of the word choices within sentences are confusing. I am going to give two examples:
o In the third paragraph of the introduction the authors use the word “Hence” when it seems like the second sentence does not agree with the prior sentence
“Outcome transformation is one solution to overcome the skewness. Such transformation should be straightforward and generates interpretable results. Hence, the results of a complex transformation may not be transparent and needs back calculations.”
o In the fourth paragraph it seems like the authors are stating that most statistical methods would only work when using non-negative values AND that these are non-negative values. So I am not sure how this is important when discussing linear regression models.
“However, the domain of some statistical distributions is for non-negative values, and as mentioned above, the cognitive outcomes are non-negative integers (ranged between 0 and 30).”
Study participants
· The study population is a little confusing. The authors state the participants had a diagnosis of subjective cognitive decline, which suggests it is a self-report of confusion or memory loss, or mild cognitive impairment which suggests a formal evaluation. But it is still unclear if inclusion criteria included a formal diagnosis of dementia, Alzheimer’s disease, family history, etc
Covariates or predictor variables
· Why did the authors choose these variables (i.e. clinical relevance or statsital testing)? Where other variables considered such as family history, age at diagnosis, neuroimaging findings?
· Although the authors look at covariates or predictor variables, it is unclear whether these predictor variables are helpful regardless of the underlying neurocognitive disorder.
· Are the authors concerned about using education level as a covariate or predictor variable when the MoCA score adds points based on education level?
Conclusion
· In the discussion the authors make statements that are difficult for the reviewer to reconcile. First, “On the other hand, we found inconsistent findings when testing the two cognitive outcomes using the inverse score scenarios.” And later, “The results of this study support the use of analysis of inverse cognitive scores over the original scores.”
· This could be flushed out more to make the reader assured the statistical model is appropriate.
Minor Comments
Abstract
· The first sentence of the abstract starts, “Cognitive for patients with…” which does not make much sense. It is possible the authors meant “Cognition for patients with...” or “Cognitive assessments for patients with...”
· It would be helpful to include the subject population in the abstract. The title indicates it is for patients with a neurocognitive disorder but no mention of that in the abstract.
Comments on the Quality of English LanguageSome of the sentence structure and word choices were confusing. See comments in the review for examples.
Author Response
Reviewer-1:
Comments and Suggestions for Authors
In the submitted manuscript, the authors evaluate several statistical models to describe the results of cognitive screening of adults at risk for neurocognitive disorders such as Alzheimer’s. The authors focused on two such screening tools: the Montreal Cognitive Assessment (MoCA) and the Visual Cognitive Assessment Test (VCAT) and evaluated various statistical models to evaluate the impact of cognitive scores. Overall, there are details in the manuscript that need to be clarified.
Reply: We thank the reviewer for reading through our manuscript and the great comments he provided, which we believe helps us improve our paper. Below, please find a point-by-point replies to the comments.
General Comments:
Contradictions or grammatical errors?
Some of the word choices within sentences are confusing. I am going to give two examples:
Reply: We thank the reviewer’s comment and have gone through our paper and revised accordingly to avoid such confusions.
- In the third paragraph of the introduction the authors use the word “Hence” when it seems like the second sentence does not agree with the prior sentence
“Outcome transformation is one solution to overcome the skewness. Such transformation should be straightforward and generates interpretable results. Hence, the results of a complex transformation may not be transparent and needs back calculations.”
Reply: We agree with reviewer’s comment and change the “Hence” word to “However”, to make it readable.
- In the fourth paragraph it seems like the authors are stating that most statistical methods would only work when using non-negative values AND that these are non-negative values. So I am not sure how this is important when discussing linear regression models.
“However, the domain of some statistical distributions is for non-negative values, and as mentioned above, the cognitive outcomes are non-negative integers (ranged between 0 and 30).”
Reply: we thank the reviewer for raising this point. We revised this section to make it easier to understand: “To counter the limitations of the normal distribution in linear regression model which can potentially provide predicted negative values, some alternative statistical distributions may be worth considering overcome this problem; and eventually improve the model performance and predictions. However, the domain of some statistical distributions is for non-negative values, and as mentioned above, the cognitive outcomes are non-negative integers (ranged between 0 and 30) which seems to fit such non-negative outcomes more appropriately.”
Study participants
- The study population is a little confusing. The authors state the participants had a diagnosis of subjective cognitive decline, which suggests it is a self-report of confusion or memory loss, or mild cognitive impairment which suggests a formal evaluation. But it is still unclear if inclusion criteria included a formal diagnosis of dementia, Alzheimer’s disease, family history, etc
Reply: we elaborated on the above point to make it easier to follow: “Participants with MCI had symptoms of cognitive impairment and objective cognitive deficits on testing, however they remained functionally intact as per the Diagnostic and Statistical Manual of Mental Disorders, Fifth Edition (DSM-5) criteria [20] or the National Institute on Aging-Alzheimer's Association (NIA-AA) criteria [21].”.
Covariates or predictor variables
- Why did the authors choose these variables (i.e. clinical relevance or statsital testing)? Where other variables considered such as family history, age at diagnosis, neuroimaging findings?
Reply: three variables (age, gender and education) were selected based on clinical relevance and as commonly used variables (as confounders) while modelling cognitive outcomes in literature. The objective of this manuscript is to study the distribution of cognitive outcomes by including no variable (intercept-only model) and 3 basic variables (age, gender and education) to elaborate on the distributional assumptions of each statistical model and the proposed inverse-transformation approach. Of course including additional variables such as family history or MRI data or even blood/genetics biomarkers will improve the model performance; however the conclusions and recommendations will remain the same. Also including many variables in the model will be subject to variable selection approach which is not within the scope of this paper, because such variable selection criteria (such as backward, stepwise etc.) will result in different list of variables for each model. For example, backward variable selection approach may come up with a different list of variables when using NB model vs GP model on the inverse VCAT, and the variable lists will vary while dealing with original VCAT. We discussed this point in Discussion: “The three predictor variables (age, gender and education) were selected as they are clinically relevant predictors of cognitive outcomes.”, “For simplicity, other variables (such as family history, neuroimaging findings, etc.) are not included in this paper, however the conclusion in terms of the choice of the model distribution and inverse-transformation will remain the same.”.
- Although the authors look at covariates or predictor variables, it is unclear whether these predictor variables are helpful regardless of the underlying neurocognitive disorder.
Reply: The predictor variables included in this study (age, gender, education) are clinically relevant predictors for cognitive outcomes. The “intercept-only model” discussed in this paper is a scenario where there is no variable in the model. So even if the three predictor variables are not helpful, the model will reduce to an intercept-only model, as if there is no variable in the model. On the second part of the comment (regardless of the underlying neurocognitive disorder): let’s say the study population is a cognitively normal cohort as a worst-case scenario (no neurocognitive disorder). Under such scenario, the conclusion will be still valid because the distribution of the cognitive outcome variables will be similar to a SCI/MCI cohort, and slightly shifted to the right side (better cognition), i.e., the skewness issue will be still there.
- Are the authors concerned about using education level as a covariate or predictor variable when the MoCA score adds points based on education level?
Reply: The MoCA (and VCAT) score in our data is not adjusted for education level, and we used the total scores. For such “education adjusted MoCA” outcome, where one point is added for those with <=12 years of education, the impact of education as a predictor may be less significant (statistically) as education is somehow within the outcome calculation. However, “education adjusted MoCA” is not within the scope of this paper. So we feel confident that including education as a predictor for MoCA (total score) is fine. We thank the reviewer for this valid point and we clarified this point under the Discussion section: “It should be noted that the cognitive outcomes are the total raw scores and not adjusted for the age or education level.”
Conclusion
- In the discussion the authors make statements that are difficult for the reviewer to reconcile. First, “On the other hand, we found inconsistent findings when testing the two cognitive outcomes using the inverse score scenarios.” And later, “The results of this study support the use of analysis of inverse cognitive scores over the original scores.”
- This could be flushed out more to make the reader assured the statistical model is appropriate.
Reply: thank you for raising this point. We revised this portion accordingly: "On the other hand, we found inconsistent findings when testing the two cognitive out-comes using the inverse score scenarios in terms of the significance of the predictor variables.", "Hence, a wrong assumed distribution for cognitive outcomes may under- or over-estimate the effect size of the predictors or the statistical significance.", "The results of this study support the use of analysis of inverse cognitive scores over the original scores, when comparing the results of inverse cognitive scores versus the original scores in terms of the model performance of each distribution."
Minor Comments
Abstract
- The first sentence of the abstract starts, “Cognitive for patients with…” which does not make much sense. It is possible the authors meant “Cognition for patients with...” or “Cognitive assessments for patients with...”
Reply: we revised this sentence and it reads as: “Cognitive assessment for patients with neurocognitive…”
- It would be helpful to include the subject population in the abstract. The title indicates it is for patients with a neurocognitive disorder but no mention of that in the abstract.
Reply: we included the study design and the subject population under the Abstract-Material and Methods.
Comments on the Quality of English Language
Some of the sentence structure and word choices were confusing. See comments in the review for examples.
Reply: we thank the reviewer for raising some of the confusing sentences. We read through our paper carefully and improved the English accordingly.
Reviewer 2 Report
Comments and Suggestions for Authors
The manuscript by Saffari, S.E. and colleagues describes statistical modeling of inverse-transformed cognitive outcomes (VCAT) in order to address the left-skewed nature of such scores. Authors have applied various statistical distribution to the transformed data under various data-analysis scenarios to assess the robustness of inverse-transformed methods and to compare them with conventional methods. The authors conclude that their data support use of inverse-transformed methods over conventional methods to model the cognitive outcomes in early stage neurodegenerative disorders.
It is an interesting study. However, the methodology is not fully explained and data is not well represented. For example, it is not clear what mathematical equations were used to obtain the modal distributions shown in Fig 2, 3, 4. What value of intercept and what coefficients in case of “with covariates” analysis were used is not clear either.
Fig 1 does not have panel marking (such as Fig 1a, b etc). The source of Fig 1 or of data therein is not specified. Also, it is not clear in Fig1 what red and blue vertical lines mean.
Fig 2,3,4, the line colors are not very distinct and therefore hard for reader to decipher which curve belong to which distribution.
Moreover, y-axis in Fig 2,3,4 says “predicted value” which is vague. A secondary y-axis is needed to indicate the observed values.
In Fig 2, none of the models seem to be fitting the observed curve. Is this due to sub-optimal fitting?
Table 6 does not show male gender beta values despite its mention in line 246.
Supplemental table and figure mentioned in Section 3.6 were not provided with the manuscript. Hence, it is not possible to comment on these findings.
Instead of relying on inverse-transformed data, authors could have used a hybrid distribution to model the data. Why did authors not consider this approach?
Also, the entire manuscript is based on just one cohort. How can authors be sure that the their conclusions about inverse-transformed outcomes have general applicability?
Comments on the Quality of English Language
Linking verbs are missing in some of the sentences in the manuscript.
Reviewer 3 Report
Comments and Suggestions for Authors
Modelling the distribution of cognitive outcomes for early stage neurocognitive disorders: a model comparison approach
This study investigates compares different statistical models for analyzing cognitive screening scores (VCAT and MoCA scores) in patients with early-stage neurocognitive disorders. The conventional linear regression model assumes normality of residuals, but cognitive scores are often skewed. So, the authors test other models like log-normal, Gamma, Weibull etc. They find that Weibull and generalized Poisson (GP) models best fit the original and inverted VCAT scores respectively. The findings are validated with MoCA scores. Bootstrapping improves precision of estimates. Using the inverted scores leads to models that fit better than using original scores. Some concerns need to be addressed or clarified before the manuscript can be considered for publication in the journal.
Major comments:
1. Were any model diagnostics like residual plots done to assess fit? How did the residual distributions look?
2. Why were more complex zero-inflated or hurdle models not tested? Could they capture the distribution better?
3. How was the education variable coded and adjusted? Did it have a linear relationship with outcomes?
4. Were there any issues with convergence or computational problems while fitting some complex models?
5. Can the authors provide some practical suggestions for analysts on best practices for modelling skewed cognitive screening data?
6. Were there any major differences between the MoCA and VCAT models in terms of predictors or distributional shape?
7. What impact could the choice of model have on inferences about predictors or rates of cognitive decline over time?
8. Were any simulations done to test robustness of different model assumptions or compare power/Type I error?
Round 2
Reviewer 2 Report
Comments and Suggestions for Authors
The revised manuscript and accompanying authors' rebuttal adequately addresses all the concerns raised by this reviewer.
Author Response
We thank the reviewer for providing us great comments which helped improve our manuscript.